# Long-Term Treatment with Cannabidiol-Enriched *Cannabis* Extract Induces Synaptic Changes in the Adolescent Rat Hippocampus

**DOI:** 10.3390/ijms241411775

**Published:** 2023-07-21

**Authors:** Andrey F. L. Aguiar, Raquel M. P. Campos, Alinny R. Isaac, Yolanda Paes-Colli, Virgínia M. Carvalho, Luzia S. Sampaio, Ricardo A. de Melo Reis

**Affiliations:** 1Institute of Biophysics Carlos Chagas Filho, Federal University of Rio de Janeiro, Rio de Janeiro 21941902, Brazil; andreyaguiar@biof.ufrj.br (A.F.L.A.); camposrp@biof.ufrj.br (R.M.P.C.); yolanda@biof.ufrj.br (Y.P.-C.); ramreis@biof.ufrj.br (R.A.d.M.R.); 2Institute of Medical Biochemistry Leopoldo de Meis, Federal University of Rio de Janeiro, Rio de Janeiro 21941902, Brazil; alinnyisaac@biof.ufrj.br; 3Faculty of Pharmacy, Federal University of Rio de Janeiro, Rio de Janeiro 21941902, Brazil; virginiamc@pharma.ufrj.br

**Keywords:** *Cannabis*, endocannabinoid system, synaptic changes

## Abstract

The endocannabinoid system (eCS) is widely distributed in mammalian tissues and it is classically formed by cannabinoid receptors, endogenous bioactive lipids and its synthesis and degradation enzymes. Due to the modulatory role of eCS in synaptic activity in the Central Nervous System (CNS), phytocannabinoids have been increasingly used for the treatment of neurological disorders, even though little is known in terms of the long-term effect of these treatments on CNS development, mainly in the timeframe that comprises childhood and adolescence. Furthermore, an increased number of clinical trials using full-spectrum *Cannabis* extracts has been seen, rather than the isolated form of phytocannabinoids, when exploring the therapeutical benefits of the *Cannabis* plant. Thus, this study aims to evaluate the effect of cannabidiol (CBD)-enriched *Cannabis* extract on synaptic components in the hippocampus of rats from adolescence to early adulthood (postnatal day 45 to 60). Oral treatment of healthy male Wistar rats with a CBD-enriched *Cannabis* extract (3 mg/kg/day CBD) during 15 days did not affect food intake and water balance. There was also no negative impact on locomotor behaviour and cognitive performance. However, the hippocampal protein levels of GluA1 and GFAP were reduced in animals treated with the extract, whilst PSD95 levels were increased, which suggests rearrangement of glutamatergic synapses and modulation of astrocytic features. Microglial complexity was reduced in CA1 and CA3 regions, but no alterations in their phagocytic activity have been identified by Iba-1 and LAMP2 co-localization. Collectively, our data suggest that CBD-enriched *Cannabis* treatment may be safe and well-tolerated in healthy subjects, besides acting as a neuroprotective agent against hippocampal alterations related to the pathogenesis of excitatory and astrogliosis-mediated disorders in CNS.

## 1. Introduction

*Cannabis sativa* is a millenary plant that has been used for therapeutical purpose since the time of Ancient China [1]. Even though more than 600 secondary metabolites have already been described in the *Cannabis* composition, which comprises mainly phytocannabinoids, terpenes and flavonoids [2], the phytocannabinoids ∆9-tetrahydrocannabinol (THC) and CBD remain the most studied compounds due to their impact on human health. THC plays the role of a partial agonist of cannabinoid receptors, which may induce the cannabinoid tetrad (sedation, catalepsy, antinociception and hypothermia) and the psychoactive property of *Cannabis* [3,4]. Inversely, CBD is a multi-target molecule that does not elicit psychoactive activity and acts not only on the eCB system, but also on other targets, such as the agonist for the serotoninergic 5-HT1A receptor and the transient receptor potential vanilloid subtype 1 (TRPV1), the inhibitor of the enzyme fatty acid amide hydrolase (FAAH) and the antagonist of the G protein-coupled receptor 55 (GPR55) [5], in addition to other pharmacodynamic actions. Furthermore, CBD is more frequently associated to anti-inflammatory properties and antiepileptic activity [6,7,8]. The main target of phytocannabinoids is the endocannabinoid (eCB) system, which is highly conserved among vertebrates and is composed of cannabinoid receptors, bioactive lipids called endocannabinoids and the enzymes responsible for their synthesis and degradation on demand [9,10]. In the central nervous system (CNS), the eCB system has a key role in the modulation of excitatory and inhibitory neuronal activity via the blockade of pre-synaptic voltage-gated calcium channels, leading to a decrease in the release of neurotransmitter vesicles [11]. It also inhibits the adenylate–cyclase activity, which regulates the signalling pathways involved in synaptic maturation and plasticity in CNS [12]. In addition, glial cells are also associated to neuroplasticity and express eCB system components. For example, it has been suggested that CB1-dependent signalling improves oligodendrocyte progenitor survival and their remyelinating properties and arborization [13,14]. CB1 receptor activation in astrocytes leads to an increase in glucose oxidation and the synthesis of phospholipids and glycogen, besides induction of ketogenesis [15,16]. Furthermore, it has been shown that neuronal activity in the pedunculopontine nucleus may elicit astrocytic calcium waves in a CB1-dependent manner, culminating in gliotransmission and consequent modulation of neuronal metabotropic glutamate receptors, thus corroborating the intertwined correlation between neurons and glia [17]. In terms of microglial cells, they abundantly express the CB2 receptor, whose activation has been reported to reduce the generation of proinflammatory cytokines and chemokines, increase neuroprotection to excitotoxic stimuli and prevent blood–brain barrier damage [18,19,20].

In spite of anecdotal reports of the improved quality of life in numerous patients after Cannabis use for the treatment of seizures, pain and anxiety, the precise mechanisms by which phytocannabinoids exert their therapeutical properties had not been fully elucidated until the isolation of these compounds and description of their signalling pathways [21]. Furthermore, robust evidence in the clinical literature has shown therapeutic benefits in children and adolescents undergoing *Cannabis*-based treatment for disorders associated to increased excitatory activity and neuroinflammation, such as epilepsy and autism spectrum disorder (ASD) [22,23,24,25,26,27].

Adolescence is a critical period for neurodevelopment due to morphological and functional changes that underlie the maturation of the neural circuitry [28]. The eCB system undergoes many changes in terms of component expression during adolescence, which makes this timeframe sensitive to environmental factors that may influence the signalling mediated by eCBs [29]. So far, preclinical studies have been focusing mainly on the use of purified CBD or THC for treating different pathological conditions in animals and in vitro models, although this approach does not take into consideration the wide range of phytocannabinoids and other *Cannabis*-derived secondary metabolites that may exert therapeutic potential following administration of full-spectrum *Cannabis* extracts [30,31].

The entourage effect, a concept that relates the full-spectrum *Cannabis* oil performing better than isolated phytocannabinoids, is also considered relevant to explain potentially increased efficacy of full-spectrum extracts in the treatment of many pathologies, as reviewed elsewhere [32,33,34]. To exemplify, terpenes and minor phytocannabinoids, such as β-caryophyllene and cannabidivarin, have pharmacological properties on their own as anticonvulsant and adjunctive therapy for ASD [35,36]. Despite the increasing evidence that indicates therapeutical benefits from CBD-rich extracts [37,38], little is known in terms of long-term exposure to these compounds in such sensitive timeframe as the transition from adolescence to early adulthood in the context of therapeutics.

Altogether, these premises point to the urgency of further investigation into the effects of *Cannabis*-based treatments in the neurobiology of healthy animal models in order to identify the intrinsic role of this therapeutic approach on neuroplasticity and possible developmental impairment. Therefore, this study aimed to evaluate the molecular and cellular changes in hippocampus induced by long-term exposure to CBD-enriched *Cannabis* extract (20:1 CBD/THC ratio) during the period ranging from adolescence to early adulthood in Wistar rats, focusing on synaptic processes and glial modulation. Additionally, we also investigated possible alterations on locomotor activity and memory consolidation, which are relevant in terms of animal growth and maturation during this critical period of development.

## 2. Results

### 2.1. CBD-Enriched Extract Has No Negative Impact on Animal Growth from Adolescence to Early Adulthood

In order to assess the overall effect of a CBD-enriched extract in physiological parameters associated to animal well-being and growth, we initially evaluated urine output plus food and water intake, in addition to weight gain. In terms of body weight, two-way ANOVA showed a significant main effect of Day (F(3, 108) = 687, *p* < 0.001), but there was no effect related to Treatment (F(1, 36) = 1.26, *p* = 0.269) (Figure 1A). By the end of treatment period, animals reached the mean weight of 249.1 ± 5.7 g in Control and 258.6 ± 6.5 g in Extract groups.

Total food intake was measured prior to treatment with the CBD-enriched extract, followed by the same evaluation after 15 days. In spite of the increase in body weight during the 15-day period of the experiment, the food intake was not impacted by either Day (F(1, 17) = 0.65, *p* = 0.431) or Treatment (F(1, 17) = 1.42, *p* = 0.251) (Figure 1B). From day 0 to day 15, the food consumed changed from 24.11 ± 0.92 g to 22.56 ± 1.16 g in Control and from 23.80 ± 1.95 g to 27.60 ± 2.24 g in Extract groups.

In terms of water balance, the amount of water consumed and the volume of urine output before and after CBD-enriched extract treatment was measured. During the baseline period prior to the start of treatment, the animals ingested water in the volume of 37.14 ± 2.20 mL in Control and 34.22 ± 2.58 mL in Extract (Figure 1C) groups, whilst urine output in the course of 24 h was 9.75 ± 1.30 mL in Control and 10.60 ± 0.82 mL in Extract groups (Figure 1D). Following CBD-enriched extract treatment, there was no effect observed on water intake due to variables Day (F(1, 14) = 0.015, *p* = 0.9023) and Treatment (F(1, 14) = 0.058, *p* = 0.812), which accounted for a total volume of 35.14 ± 1.06 mL in Control and 36.67 ± 2.74 mL in Extract groups. Similarly, the volume of urine output was not affected either by Day (F(1, 16) = 8.08, *p* = 0.011) or Treatment (F(1, 16) = 0.022, *p* = 0.884), reaching mean volumes of 13.91 ± 1.24 mL in Control and 13.35 ± 1.07 mL in Extract groups. Therefore, we did not identify any negative effects induced by CBD-enriched extract treatment in parameters associated to animal growth from adolescence to young adulthood.

### 2.2. CBD-Enriched Extracts Do Not Induce Hypolocomotion and Short-Term Memory Impairment in Healthy Rats

Considering that the Cannabis extract used in this study had THC 1 mg/mL in its composition, even though it was CBD-enriched, a long-term exposure to this treatment could possibly induce CB1-dependent hypolocomotion. Therefore, an open field test was carried out, and total distance travelled by animals was assessed during 5 min. For Control, the total distance was 9.19 ± 1.17 m, whilst the Extract group travelled 8.42 ± 0.31 m (Figure 2A), and no statistical difference was identified (t (11.36) = 0.635, *p* = 0.537).

Moreover, it was also important to evaluate the impact of the eCB system modulation in the declarative memory of the animals exposed to the CBD-enriched extract due to the correlation of the eCB system and plasticity in neural circuitry, which is essential for learning and memory consolidation [39]. Therefore, short-term recognition memory was measured by the quantification of the novel object recognition index. In this experiment, the Control group showed a recognition index of 0.57 ± 0.05 a.u., whilst the index in the Extract group was 0.68 ± 0.03 a.u. (Figure 2B). In spite of a slightly higher index in the Extract group (t (18) = 1.983, *p* = 0.062), it was not possible to infer whether CBD-enriched extracts improved declarative memory. Therefore, it is safe to state that there was no short-term memory impairment induced by this CBD-enriched extract treatment.

### 2.3. CBD-Enriched Extract Modulates Protein Levels of Synaptic Plasticity and Astrogliosis Markers in Hippocampus

In CNS, the eCB system is involved in controlling several biological processes at the synaptic cleft, such as the release of excitatory and inhibitory neurotransmitters and ultimately protein synthesis. Therefore, the levels of protein markers associated to glutamatergic synapses and astrogliosis were measured considering that these are the two main features affected in pathological scenarios such as epilepsy and autism spectrum disorder, which are more commonly diagnosed in children and adolescents under treatment with Cannabis-based products (Figure 3).

In spite of the presence of a 1 mg/mL THC in its composition, the CBD-enriched extract did not induce changes in protein levels of the CB1 receptor in the hippocampus (0.71 ± 0.08 a.u. in Control and 0.62 ± 0.10 a.u. in Extract groups, t (20.96) = 0.759, *p* = 0.456), as shown in Figure 3B. Similarly, for CB2, which is more commonly found in microglial cells, receptor levels were also not affected by treatment (1.88 ± 0.22 a.u. in Control and 1.57 ± 0.15 a.u. in Extract groups, t (21.42) = 1.178, *p* = 0.251), shown in Figure 3B. Even though healthy animals were used in this study, the GFAP protein level, which is used as a marker of astrocyte presence in order to infer astrogliosis, was 0.52 ± 0.05 a.u. in Control and 0.36 ± 0.03 a.u. in Extract groups (t (31.04) = 2.717, *p* = 0.010), which accounts for a reduction of almost 30% in animals treated with the CBD-enriched extract (Figure 3D).

Considering glutamatergic synapses, we focused on two subunits of the main ionotropic receptors, recently implicated in paediatric neurological disorders for which Cannabis-based medicines have been more often prescribed. The expression of the GluA1 subunit of the AMPA receptor was reduced in 29% in the hippocampus of extract-treated rats (0.78 ± 0.06 a.u. in Control and 0.56 ± 0.05 a.u. in Extract groups; t (33.92) = 3.035, *p* = 0.004; Figure 3E), whereas the amount of the GluN1 subunit of the NMDA receptor was not different upon treatment (0.65 ± 0.07 a.u. in Control and 0.75 ± 0.19 a.u. in Extract groups; t (20.38) = 0.8260, *p* = 0.418; Figure 3F).

Additionally, the level of scaffold proteins related to the stability of dendritic spines and neurotransmitter release was evaluated. Results showed an increase of 42% in the PSD95 level compared to control (0.52 ± 0.03 a.u. in Control and 0.74 ± 0.08 a.u. in Extract grous, t (22.09) = 2.534, *p* = 0.018), suggesting a modulation of post-synaptic component in the CBD-mediated effect (Figure 3G). Comparatively, the levels of SYN, which account for the pre-synaptic component of synapses by mediating the fusion of neurotransmitter vesicles at the synaptic cleft, remained unaffected after treatment with the CBD-enriched extract (1.18 ± 0.06 a.u. in Control and 1.14 ± 0.07 a.u. in Extract groups, t (30.55) = 0.4311, *p* = 0.669; Figure 3H).

### 2.4. GFAP-Positive Staining in Hippocampus Was Not Altered after Treatment

Considering the reduction observed in GFAP levels in total hippocampus homogenates evaluated by Western blotting, we further investigated the tissue distribution of GFAP-positive cells. We focused our analysis on glial cells due to their expression of cannabinoid receptors and reports in the literature that show modulation in the activation state/gliosis after exposure to CB1/2 agonists, as previously published [40,41,42]; therefore, immunostaining for PSD95 was not performed. The presence of cannabinoid receptors in astrocytes and their involvement in neuroinflammatory pathways, besides their role regulating gliotransmission and synaptic plasticity, raises the relevance of this cell type in the context of *Cannabis*-based treatments.

In order to perform such analysis, we measured the percentage of the GFAP-positive area relative to the whole hippocampus in brain slices. In spite of the difference between both groups that have been demonstrated by Western blotting, we showed that 9.39 ± 0.59% of Control group hippocampi were stained for GFAP, similarly to the area in the Extract group that reached 8.17 ± 1.49% (*U* = 4, *p* = 0.190; Figure 4I). In Figure 4C–H, we show the inset of the CA1 region in higher magnification to provide an insight in terms of staining pattern of GFAP-positive cells.

### 2.5. CBD-Enriched Extracts Reduce Microglial Cell Complexity in Subregions of Hippocampus

To further understand how CBD-enriched extracts could alter the neural functioning in healthy animals, immunohistochemical analysis of microglia morphology was performed using the marker Ionized calcium-binding adapter molecule 1 (Iba-1), followed by imaging processing and Sholl analysis to infer cell morphological complexity in regions CA1, CA3 and dentate gyrus (DG) of the hippocampus. Morphological changes in microglia induced by treatment were expressed as area under curve from Sholl XY graph plotting, which therefore expresses the complexity level of microglia morphology. Thus, higher area under the curve indicates more arborized cells.

We observed a tendency of decrease in the area under the curve of Sholl-analysed cells after treatment in CA1 (41.33 ± 3.29 a.u. in Control and 30.33 ± 2.59 a.u. in Extract groups; t (4.184) = 2.629, *p* = 0.056), as seen in Figure 5C,F. However, in CA3, there was an approximately 40% reduction in microglial complexity (44.08 ± 3.43 a.u. in Control and 26.41 ± 2.00 a.u. in Extract groups; t (3.334) = 4.445, *p* = 0.016), possibly related to a decreased number of microglial processes closer to soma (Figure 5D,G). Differently from what was observed in other hippocampal regions, there was no treatment-induced change in microglial morphology in DG (29.81 ± 1.50 a.u. in Control and 29.10 ± 2.11 a.u. in Extract groups; t (4.912) = 0.2744, *p* = 0.794).

### 2.6. Phagocytic Activity of Microglial Cells Was Not Modulated by CBD-Enriched Extract

In CNS, microglial cells are essential for surveillance and synaptic pruning by engulfing dendritic spines and to promote maturation of synapses, which is critical for neurodevelopment. Considering that our data showed reduction in the GluA1 protein level in the hippocampus, we raised the hypothesis that microglial phagocytic activity could have been modulated by the CBD-enriched extract, thus changing the dynamics of the AMPA receptor in dendritic spines. To elucidate this aspect, hippocampal slices were immunostained for Iba-1 and LAMP2 in order to quantify the number of phagocytic vacuoles inside microglial cells, which would be used to infer the possible role of microglia in the reduced levels of GluA1. Alongside LAMP1, LAMP2 is a major protein component of lysosomal vacuoles, in which structures are degraded after phagocytosis [43]. Thus, the number of LAMP2-positive organelles in microglia can be safely used as a marker for phagocytic activity.

The total number of Iba-1 and LAMP2 co-marked events was quantified and revealed that, despite morphological changes in microglial cells, the phagocytic activity of these cells was not modulated by the CBD-enriched extract in CA1 (9.77 ± 1.11 events in Control and 10.40 ± 2.08 events in Extract groups; t (6.111) = 0.2682, *p* = 0.797; Figure 6B), CA3 (9.60 ± 1.30 events in Control and 10.37 ± 1.43 events in Extract groups; t (7.930) = 0.3975, *p* = 0.701; Figure 6C) and DG (9.27 ± 1.87 events in Control and 10.03 ± 1.91 events in Extract groups; t (7.997) = 0.2866, *p* = 0.781,; Figure 6D).

## 3. Discussion

This study described the long-term effects of treatment with the CBD-enriched *Cannabis* extract on the behaviour and the neural cells of healthy male rats in the timeframe of adolescence to early adulthood (PND 45–60). We focused on the cellular and molecular changes to the synaptic structure and glial reactivity, further exploring the correlation of these alterations to behavioural aspects associated to locomotion and cognition. In general, our results showed that CBD-enriched extracts are able to modulate the molecular composition of glutamatergic synapses, GFAP expression in astrocytes and morphology of microglial cells, even in the absence of behavioural alterations.

The transition from adolescence to early adulthood is a critical period for brain development due to the remodelling of morphological and functional features to promote brain maturation [28]. Considering that the use of *Cannabis*-based products for the treatment of neurological disorders that affect children is still in its infancy and the number of paediatric patients undergoing these courses of treatment has been increasing [44,45], it is of the utmost importance to better understand how these compounds may affect normal brain development. Moreover, the majority of studies with phytocannabinoids focus on isolated molecules instead of the synergistic potential of their combination, which has been increasingly added to the treatment regimen of patients worldwide. Thus, the collective contribution of compounds found in *Cannabis* towards a better clinical outcome may be relevant to account for potentiated effects in animals and humans. In fact, a recent meta-analysis has shown that CBD-rich extracts are more efficient and are safer than isolated CBD in patients with treatment-resistant epilepsy, mostly children and adolescents [46].

It is well-known that cannabinoid receptor agonists have the capability to induce feeding through the increase in hypothalamic POMC neuron activity by a mitochondria-mediated mechanism [47]. In addition, CB1 knockout and intracerebroventricular injections of CB1-selective antagonist promote reduction in food intake in animals [48,49]. For paediatric patients who often rely on ketogenic diet or other dietary restrictions to reduce the phenotypic outcomes of neurological disorders, such as ASD and epilepsy [50,51], changes to feeding patterns and water balance might be key factors to disregard *Cannabis*-based products in their treatment regimen. Our results showed that CBD-enriched extracts, despite the presence of low THC concentration, do not influence the dietary behaviour of rats, therefore not affecting their body weight, food and water consumption.

THC is the psychotropic component of *Cannabis* and its pharmacological effect is promoted by its binding to the CB1 receptor, therefore inducing hypolocomotion, catalepsy, hypothermia and analgesia, which altogether are referred to as the cannabinoid tetrad [52]. In spite of the pharmacokinetic benefit of THC and CBD co-administration, which increases the plasmatic bioavailability of both compounds and allows lower dosage of full-spectrum *Cannabis* products in comparison to isolated CBD [53], it is relevant to assess the detrimental effect of exogenous CB1 modulators during the transition from adolescence to early adulthood. Thus, we performed the open field test to measure locomotor activity, and no changes were observed in animals treated with the CBD-enriched *Cannabis* extract, therefore suggesting that the daily THC dose below 0.16 mg/kg/day used in our protocol was not enough to evocate the behavioural features associated to agonistic activity on CB1 receptors.

In terms of cognitive development, the physiological activation of cannabinoid receptors plays the role of circuit-breaker, acting as a retrograde modulator of synaptic activity that may exert its effect by two main pathways: a short-term modulation mediated by the direct blockade of presynaptic voltage-gated calcium channels [11,54], and a long-term effect of the CB1 receptor associated to the inhibition of adenylyl cyclase and suppression of protein kinase A activity by the G-protein αi/o limb, further mediating the process known as long-term depression (LTD) [55]. It has been shown that WIN 55212-2, a non-selective cannabinoid agonist, impairs contextual fear conditioning in Wistar rats in a CB1-dependent manner [56]. Additionally, the JWH-018 agonist and its halogenated derivatives, besides ∆9-THC, negatively impacted short- and long-term memory consolidation in the NOR test in mice [57].

Considering these aspects related to CB1 activation and the neurological basis for memory formation, early exposure to THC in full-spectrum CBD-enriched *Cannabis* extracts may be harmful to the developing brain of children. Thus, we used the NOR test to assess short-term memory following the 15-day period of treatment. Surprisingly, we found a tendency of improved performance in extract-treated animals. Whilst acute administration of CBD-rich *Cannabis* extract had no impact on the working memory of rats [58], it has been shown that chronic treatment with CBD is able to restore the social and object recognition, working memory and social interaction deficits in the young rat model of prenatal infection by poli I:C and transgenic APPswe/PS1∆E9 mice [59,60]. A similar outcome was also found in 14-month-old female TAU58/2 transgenic mice, which have phenotypic features associated to frontotemporal dementia and Alzheimer’s disease [61]. Therefore, a possible improvement in memory performance could be explained by the long-term exposure to the CBD-enriched extract, whilst THC concentration may not be enough to elicit the CB1-dependent cognitive impairment.

In order to further expand our observations into the molecular aspect of synapses, we investigated the expression of proteins associated to the eCB signalling and the components of glutamatergic synapses, which are highly correlated to the pathogenesis of disorders such as epilepsy and ASD [62,63]. We focused our study on the hippocampus due to its relevance for processing of memory and learning, which is pivotal for a developing individual, the presence of a neurogenic site located at the dentate gyrus and its functional correlation and integration with the prefrontal cortex during maturation [64,65]. In fact, it has been reported that acute coadministration of CBD 50 mg/kg and THC 1 mg/kg upregulates the expression of the CB1 receptor in the hippocampus and hypothalamus [66]. Moreover, the effect of long-term exposure to THC- and CBD-rich extracts throughout adolescence has already been studied in female rats, and it was shown that the CBD-enriched extract reverts the THC-induced downregulation of the CB1 receptor in the prefrontal cortex [67]. In the fear-conditioning paradigm, aversive memory reconsolidation has been disrupted after CBD treatment in a CB1-dependent manner, even though the contribution of CB2 receptors is still controversial [68,69,70]. We showed that the expression of CB1 and CB2 receptors have not been modulated by the CBD-enriched extract in healthy rats, which may suggest that either the CBD/THC ratio of approximately 20:1 or the dose of both phytocannabinoids used prevented the alteration in the CB1 and CB2 expression reported in other studies.

As mentioned, the cannabinoid-mediated retrograde signalling modulates neurotransmitter release, thus locally controlling synaptic activity. Even though cannabinoid receptors are widely distributed in the brain, its occurrence in glutamatergic neurons is necessary for on-demand protection in pathological contexts of exacerbated excitatory activity [71,72]. Therefore, we evaluated the expression of proteins associated to glutamatergic signalling. Focusing on subunits of ionotropic glutamate receptors, we showed that the GluA1 subunit of the AMPA receptor is downregulated in animals treated with the CBD-enriched extract, but no changes were seen regarding the GluN1 subunit of the NMDA receptor. In fact, the murine model of Rett syndrome by Mecp2 knockout shows a disrupted formation of LTP and an altered rectification of evoked and quantal AMPAR-mediated excitatory postsynaptic currents (EPSCs), accompanied by an elevation of GluA1 protein levels in the hippocampus [73]. In addition to subcortical regions, it has also been shown that GluA1 is upregulated in the prefrontal cortex of in utero valproic acid-exposed animals, thus pointing to the participation of an AMPA-mediated unbalance on the behavioural impairment in ASD models [74]. Interestingly, CBD actively modulates the electrophysiological properties of GluA1 through the interaction with its N-terminal domain, which leads to faster deactivation and slower recovery of this subunit, thus reducing AMPAR-mediated EPCSs in the febrile seizure model [75]. Our findings suggest that CBD-enriched extracts may promote neuroprotective adaptations on the expression of GluA1, thus contributing to the decreased impact of excitotoxicity on behavioural changes.

Apart from the short-term action of cannabinoids, further activity-dependent structural changes to synapses are underlying the modulation of synaptic efficacy [76]. Considering the reduction in GluA1 protein levels, we hypothesised that CBD-enriched extracts might be shifting the dynamic of dendritic spines and therefore the mechanisms of synaptic plasticity. The correlation between AMPAR and PSD95 has already been studied and it has been shown that overexpression of PSD95 in cortical pyramidal neurons increases the amplitude of AMPAR-mediated evoked EPSCs, in addition to a higher frequency of AMPAR-mediated miniature EPSCs [77]. Furthermore, impaired stability of dendritic spines, in which PSD95 is a pivotal scaffolding protein, have been implicated in the pathophysiological features of neurological disorders, such as depression, Alzheimer’s disease and DLG4-related synaptopathy [78,79,80]. Our findings show that CBD-enriched *Cannabis* extracts are able to increase protein levels of PSD95 in the hippocampus, which suggests altered dynamics in dendritic spines of excitatory synapses in spite of GluA1’s own dynamics. In a study that used pure CBD (7–30 mg/kg i.p.) to treat mice submitted to the forced swim test, the acute antidepressant property of CBD was associated to an increased expression of SYN and PSD95 in the medial prefrontal cortex, even though there was no change in the levels of these synaptic proteins in the hippocampus [81]. Conversely, mice exposed to the chronic unpredictable stress protocol showed an increased expression of synapsin Ia/b and PSD95 in hippocampal synaptoneurosomes after treatment with CBD (10 mg/kg i.p.), alongside anxiolytic responses in the elevated-plus maze and novelty suppressed feeding tests [82]. Similar observation in terms of synaptic proteins has been made in a model of transient global cerebral ischemia in rats, which showed a decrease in ischemia-induced memory impairment, accompanied by a slight attenuation of SYN and PSD95 protein levels in the hippocampus after CBD treatment (10 mg/kg i.p.) [83]. Among the reasons for the contrasting impact of CBD treatments on PSD95 expression, we can emphasize the different species used for experiments, structural and physiological features of brain regions analysed, administration route, CBD dosage and its formulation once full-spectrum extracts have several other unknown components that may be acting on other targets.

In addition to neurons, whose expression of components associated to the excitatory machinery was evaluated, glial cells also possess all the molecular machinery required for the synthesis, transport and degradation of endocannabinoids. They also express cannabinoid receptors whose activation is highly correlated to gliotransmission and glia-mediated neuroinflammation, therefore making them a possible target for phytocannabinoids and extract-induced modulation [84]. Taking into consideration the prevalence of *Cannabis*-based medicines in children with neurological disorders characterised by neuroinflammation and excitation/inhibition unbalance, such as epilepsy and ASD, and that this age is a critical period for synapse maturation, we chose to investigate possible neurochemical and glial alterations. Thus, we focused our observations on microglial and astrocytic dynamics due to their role in the development, inflammation and establishment of pathological microenvironments, as previously reviewed by several research groups [85,86,87,88].

Astrocytes are essential for supporting the microenvironment in which synapses take place due to their activity on ionic and osmotic homeostasis, glutamate reuptake and gliotransmission. However, damages to the CNS integrity induce astrocytic reactivity that culminates in an increased expression of cytoskeleton proteins, notably GFAP, a release of several cytokines and chemokines, such as IL-10, TNFα and IL-1β, and a lessening of morphological processes, thus reducing cellular arborization [89]. Altogether, these alterations may contribute to the development and establishment of neurological disorders, even after the original insult is resolved [90]. Even in the absence of a pathogenic injury, the exacerbation of reactive astrogliosis by integrin β1 knockout induces the onset of spontaneous seizures in mice [91], which is also in alignment with the observations of elevated protein levels of GFAP in the dorsal hippocampus of mice after induction of status epilepticus by pilocarpine [92]. Hence, we investigated the expression of GFAP in hippocampi of animals treated with CBD-enriched extract, besides evaluating the tissue distribution of GFAP-positive cells. Our findings show reduction in GFAP protein levels following treatment, which is in alignment with other studies that demonstrated prevention of reactive astrogliosis following CBD administration. For example, Gáll et al. (2022) showed that the oral administration of CBD (60 mg/kg) prevented the proliferation of GFAP-positive cells in the hippocampus of the rat pentylenetetrazol-kindling model of epilepsy, in alignment with the effect of peritoneal high dosage of CBD that exerts the same decrease in astrocyte hyperplasia [93].

Furthermore, we used the immunohistochemistry technique to visually assess the result obtained by Western blotting analysis and quantified the percentage of GFAP-positive staining over the total hippocampal area, but there was no difference between the groups. One hypothesis for the divergent results obtained in both experiments may be that Western blotting offers the perspective of the whole-tissue expression of proteins, whilst immunohistochemistry focuses on the spatial distribution of cells throughout tissue sections, which may bias the observation.

Besides astrocytes, another cell type that is relevant for synaptic maturation and brain homeostasis is microglial cells, which are CNS-resident specialized macrophages derived from yolk sac peripheral mesodermal cells that infiltrate the CNS and differentiate into mature microglia [94]. According to the activation state, these cells were originally classified as the M1 or M2 phenotype since they are associated to the pro-inflammatory and anti-inflammatory profile, respectively [95]. However, more recent studies have been pointing to an activation spectrum that relies not only on morphological features, such as ramification of processes and soma roundness, but also on the variety of molecules secreted and expressed in the cytoplasmic membrane [96]. Therefore, we investigated the effect of CBD-enriched extracts on the morphological ramification of microglia in regions CA1, CA3 and DG of the hippocampus due to their expression of cannabinoid receptors, which makes them sensitive to our experimental intervention. Our data showed reduced microglial arborization in CA3 regions in the treated animals, besides elevated tendency of similar effect on CA1 region, despite not reaching the significance threshold set for experiments. Our data show that treatment with the CBD-enriched *Cannabis* extract significatively impacted microglial morphology in the CA3 region of the hippocampus. However, it is not possible to infer whether this effect is a result of direct phenotypic modulation or changes in the secretory profile of microglial cells. It has been shown that CBD and *Cannabis* extracts exert anti-inflammatory activity in a CB2-dependent mechanism, which affects the morphology and release of anti-inflammatory cytokines [97]. In addition, the classic description of microglial morphology in terms of basal state, M1 and M2 does not account for its vast repertoire of secreted factors during neurodevelopment, synaptic plasticity, aging and pathological conditions, in which the integration of epigenetic factors, transcriptomics, metabolomics and proteomics lead to a multidimensional integration of co-existing microglial states [98].

Our data also showed that CBD-enriched extracts do not modulate the phagocytic activity of microglia in the CA1, CA3 and DG regions of hippocampus. During brain development, it has been shown that microglia rely on the fractalkine-mediated signalling to engulf synaptic material from both pre- and post-synaptic terminals, thus promoting the pruning of dendritic spines [99]. At the opposite side, increased synapse elimination due to the phagocytic activity of microglial cells is associated to the pathophysiology of neurological disorders, such as demonstrated in vitro with schizophrenia-patient-derived induced microglia-like cells [100]. LAMP-2 is a protein present in the membrane of lysosomes. By quantifying the number of lysosomes inside each microglial cell in CA1, CA3 and DG, we were able to evaluate whether the phagocytic activity of microglial cells would be altered by the CBD-enriched *Cannabis* extract administration when compared to animals that received TCM [101]. There was no significant difference between the experimental groups, indicating that the treatment with the extract did not influence microglia phagocytic activity.

Recently, CBD administered twice daily (20 mg/kg, i.p.) in male and female C57BL/6J mice in the adolescent period (25–45 days) was shown to be innocuous regarding behavioural tests on motricity, anxiety, and spatial memory, compared to vehicle-treated mice [102]. Similar conclusion was described on a previous paper when CBD was administered to adult C57Bl/6J mice (20 mg/kg/day for 6 weeks, i.p.), beginning at 3 or 5 months, and the animal behaviour was not different compared to that of control animals on the influence of prolonged CBD treatment [103].

## 4. Materials and Methods

### 4.1. Cannabis Extract Preparation and Analysis

CBD-enriched *Cannabis* extract was obtained by supercritical fluid extraction (SFE) [104]. *Cannabis* flowers of Harle Tsu variety were used for the preparation of medicinal *Cannabis* extract used in the treatment of a participant attended by a medical *Cannabis* monitoring program, called the Farmacannabis project, that was approved by the Ethic Committee of the Clementino Fraga Filho Hospital, Rio de Janeiro, Brazil, Number 2021817.0.00005257, and a share was kindly donated for research purposes. The participant received judicial authorisation for *Cannabis* cultivation and the Farmacannabis project was appointed by a judge to monitor the crops and provide technical pharmaceutical support for *Cannabis* extract preparation. Dried *Cannabis* flowers were crushed in a blender and heated at 110 °C for 5 h in a forced air stove (Bio SED-CR, 85 L, 7Lab, Rio de Janeiro, Brazil). Cannabinoids were extracted by supercritical fluid extractor system (Top Industrie, Green Lab System; Vaux le Penil, France) equipped with a 1000 bar high-pressure pump, an XU032 stove (Vaux le Penil, France) and a 100 mL cell. CO_2_ gas (analytical grade, supplied by Linde PLC, Dublin, Ireland) was used as extraction solvent. The SFE system was set up: vegetable mass of 10 g, pressure of 15 MPa, cell temperature of 50 °C, CO_2_ mass of 800 g and ethanol of USP grade 50 mL (Scharlau, Barcelona, Spain), as well as ethanol of 50 mL were placed in the separator reservoir. After cannabinoid extraction, ethanol was removed by rotary vacuum evaporation. The obtained resin was diluted in medium chain triglycerides (MCT; Brain TCM^®^, Puravida, La Jolla, CA, USA) and cannabinoids CBD, THC, their acidic forms, respectively, cannabidiolic acid (CBDA) and ∆9-tetrahydrocannabinolic acid (THCA), besides the THC degradation by-product, cannabinol (CBN), were quantified by a validated High-Performance Liquid Chromatography–Photodiode Array Detector (HPLC-PDA) method, as previously published [105]. Briefly, cannabinoids were extracted via an ultrasound bath with methanol:n-hexane (9:1) and quantified by the reversed phase HPLC-PDA method. Cannabinoid concentrations were 24.0 mg/mL, 1.0 mg/mL and 0.5 mg/mL for CBD, THC and CBDA, respectively, whilst THCA and CBN were not detected.

### 4.2. Animals and Extract Administration

After weaning, male Wistar rats were housed in cages measuring 40 cm in length × 34.5 cm in width × 17 cm in height (4 to 5 animals per cage) under a 12 h light–dark cycle (dark cycle from 7:00 to 19:00) and a controlled room temperature (23 ± 1 °C), with free access to water and food. On postnatal day (PND) 45, they were divided into two groups to receive the appropriate oral solution. All procedures conducted with a total of 48 rats were approved by the Ethical Committee for the Use of Animals in Research of Federal University of Rio de Janeiro (014/19). Following random allocation to experimental groups, control animals received the oily solution of medium chain triglycerides (MCT; Brain TCM^®^, Puravida), and experimental animals were treated with the CBD-enriched *Cannabis* extract (a CBD dose of 3 mg/kg/day), both administered in the oromucosal region from PND45 to PND60. Between 15:00 and 17:00, a pipette was used to gently place the viscous solution in contact with the buccal mucosa and avoid stress from the gavage.

### 4.3. Metabolic Cage

Before administration of the extract (day 0), the animals were individually housed in metabolic cages over 24 h in order to assess their water and food intake, as well as urine flow. After the 15 day treatment period, the animals were once again kept in the metabolic cages over further 24 h and the same parameters were measured. By the end of the 24 h period in the cage (PND30 and PND60), food intake, water consumption and urine output were measured, while body weight was measured at PND45, PND50, PND55 and PND60. The animals (*n* = 9 in Control and *n* = 10 in Extract groups) that were evaluated in this parameter were not used for behavioural tests because the metal grid and the confining environment could result in stress and confound experiments. Thus, 4 animals from 5 different dams were directed to the metabolic cage.

### 4.4. Behavioural Tests

All behavioural tests were carried out in the dark phase (between 9:00 and 12:00) under undirect red light. Prior to testing, the animals were conditioned in the test room (22 ± 1 °C) for one hour and all apparatus was cleaned between the testing of animals with a 70% ethanol to remove olfactory cues. After behavioural tests were carried out, the animals (*n* = 11 for both groups) were immediately euthanised, therefore no treatment administration was performed on the same day when behavioural tests were conducted (P60).

### 4.5. Open Field

After habituation in the test room, the animals were individually placed in the centre of a cubic arena measuring 37 cm × 37 cm × 37 cm. In order to assess general locomotor activity measured by the total distance travelled in meters, the rats were allowed to freely explore the arena during 5 min [106]. Their activity was recorded with a Sony Handycam DCR-SX40 camera for further quantification of total distance travelled with the assistance of the ANY-maze software, version 7.20 (Stoelting Co., Wood Dale, IL, USA).

### 4.6. Novel Object Recognition (NOR) Test

Following a 1 h resting period after the open field test, the rats were assessed in the NOR test, which was used to evaluate the short-term recognition memory. First, they were allowed to explore 2 identical objects during 5 min (training session). Four pairs of different objects were used randomly to avoid colour and format preferences and included blue Lego towers (3 cm × 3 cm × 6 cm), orange plastic cylinders (3.8 cm diameter × 4.5 cm height), red plastic cylinders (4 cm diameter × 5.7 cm height) and pink cylinders with an octagonal base (3.5 cm × 3.5 cm × 5 cm). After 1 h, one of the familiar objects was replaced for a novel one and the test session was conducted for additional 5 min. Object interaction was defined by the animal sniffing or touching the objects with forepaws while sniffing [59], and the test was video-recorded for later manual evaluation of time spent interacting with both objects with the help of stopwatches. The exploration index was reported as the dependent variable in graphs was calculated in terms of the time spent exploring the novel object in relation to the total time exploring both objects. The objects and the arena were cleaned with a 70% alcohol in order to mitigate odour cues [107]. The animals that did not explore one of the objects during test assessment were removed from analysis.

### 4.7. Western Blotting

For molecular analysis, the animals (*n* = 19 in each group) were euthanised by decapitation at PND60 after completing the behavioural tests, and their brains were rapidly removed and washed with cold saline solution (sodium chloride 0.9%). Hippocampi were dissected on a cold surface, frozen in liquid nitrogen and conditioned at −80 °C until sample processing. For sample preparation, tissues were manually homogenised in a lysis buffer (HEPES 50 mM, MgCl_2_ 1 mM, EDTA 10 mM and Triton X-100 1%, pH 6.4) and a protease inhibitor cocktail (1:100, Sigma-Aldrich, P2714-1BTL), followed by protein quantification by the method of Lowry (1951) [108]. After running in a 12% SDS-PAGE, proteins were transferred to nitrocellulose membranes with a tris-glycine buffer (Tris-base 192 mM, glycine 25 mM and methanol 20%) and blocked with skim milk of 5% in the tris-buffered saline, followed by incubation with the following primary antibodies overnight at 4 °C: anti-CB1 receptor (1:1000, Cusabio, CSB-PA007048, Houston, TX, USA), anti-CB2 receptor (1:1000, Sigma-Aldrich, WH0001269M1, St. Louis, MO, USA), anti-GFAP (1:1000, Sigma-Aldrich, G3893), anti-synaptophysin (1:1000, Sigma-Aldrich, SAB4200544), anti-PSD95 (1:1000, Cell Signaling, D27E11, Danvers, MA, USA), anti-GluA1 (1:1000, Abcam, AB31232, Cambridge, UK), anti-GluN1 (1:1000, Abcam, AB17345) and anti-β-actin (1:10,000, Sigma-Aldrich, A5316). Afterwards, a 2 h incubation with horseradish peroxidase (HRP)-conjugated secondary antibodies at room temperature was performed using the following: HRP anti-rabbit (1:5000, Sigma-Aldrich, A0545) and HRP anti-mouse (1:5000, Sigma-Aldrich, A5278). The signal was assessed with the addition of the Immobilon Forte Western HRP substrate (Merck-Millipore, WBLUF0500, Burlington, MA, USA) and detected in a Chemidoc MP Imaging System (Bio-rad) prior to further quantification in the Image Lab software (Bio-rad). Membranes used for multiple protein analyses were stripped with glycine of 0.2 M and pH of 2.2 at 50 °C for 30 min. Ultimately, each protein content, analysed by optical density, was normalised with β-actin levels.

### 4.8. Immunohistochemistry and Image Acquisition

A group of animals different from the ones used for Western blotting analysis were euthanised in a CO_2_ chamber, the animals (*n* = 5 in each group) were perfused with sodium chloride of 0.9% and paraformaldehyde of 4% in a phosphate buffer of 0.1 M. Brains were dissected and kept in the fixative for 24 h, followed by immersion in solutions of increasing concentrations of sucrose (10%, 20% and 30%). Brain slicing was performed in a cryostat in the presence of an optimal cutting temperature solution and a mean temperature of −24 °C. The sections used for immunohistochemistry analyses were from −2.52 mm to −3.84 mm in relation to Bregma [109].

Glial cells and lysosomes from hippocampi were assessed by immunohistochemistry in coronal brain slices, 12 µm thick. Samples were washed with phosphate buffer saline (PBS) with a 0.5% Triton X-100 3 times, 5 min each, and then incubated with normal goat serum of 5% for 1 h at room temperature (RT). Primary antibodies, anti Iba-1 (1:500, Synaptic Systems, 234-013, Göttingen, Germany), anti-GFAP (1:500, Dako, Z0334, Glostrup, Denmark) and anti-LAMP2 (1:300, Santa Cruz, sc-18822, Santa Cruz, CA, USA), were incubated overnight at 4 °C. Afterwards, PBS washes were performed and secondary antibodies, goat anti-rabbit Alexa 594 (1:400, Life Technologies, A21207, Carlsbad, CA, USA) and goat anti-mouse Alexa 488 (1:400, Life Technologies, A21202), were incubated for 2 h in RT, followed by DAPI staining (1:1000; Sigma) and glycerol of 40% in PBS, pH of 7.4, was used as a coverslip sealant.

Image acquisition was carried out in the ApoTome microscope, Axio Image M.2 (Zeiss, Jena, Germany), with 20×, 40× and 63× objectives. For microglia morphology, each animal had 2 different brain slices photographed in the region of CA1, CA3 and the dentate gyrus of the hippocampus, with the 40× objective and a Z-stack accommodating the number of photos necessary to reconstruct the whole microglial cells in the field. Phagocytosis analysis was based on the images from the same hippocampus areas photographed for microglia morphology, but new images were acquired with a 63× objective. For quantification of the GFAP-positive area, 4 hippocampi from two different brain slices of each animal were photographed with a 20× objective and mosaic reconstruction, followed by processing in the ImageJ software (version 1.53t) for hippocampi delimitation and determination of percentage of the GFAP-stained area.

### 4.9. Microglia Morphology Analysis

Using the ZEN Blue 2012SP1 software (Zeiss), 3D Z-stack photos were converted into 2D by maximum-intensity projection and exported to the tiff format. Skeleton and Sholl analyses were performed using the FIJI software as previously described [101,110]. Briefly, the images were transformed to 8 bits, the threshold for Iba-1 staining was defined and the macro was generated, which was the image used for the skeleton or Sholl analysis.

### 4.10. Phagocytosis Analysis

The number of LAMP-2-positive puncta inside the Iba-1 positive cells was quantified in the ZEN Blue 2012SP1 software using the orthogonal projection in order to certify that LAMP-2 staining was inside microglial cells [101]. Images from the Iba-1 positive cells were taken with a 63 × objective using the apotome mode for the Z-stack (the number of stacks varies according to the cell, but in all cases, enough stacks were made to acquire the image from the whole of Iba-1-positive cells). The quantification of LAMP-2 positive puncta inside Iba-1 positive cells was performed manually in the ZEN Blue 2012SP1 software. Using the orthogonal projection, each stack from every Iba-1 cell was analysed in order to quantify the number of LAMP-2 puncta inside each microglial cell. This analysis was performed in all Iba-1-positive cells in the photo from CA1, CA3 and DG from each animal.

### 4.11. Statistical Analysis

The normal distribution of data was assessed by Shapiro–Wilk normality test prior to further analysis. Results from metabolic cage experiments were analysed by two-way ANOVA (Day and Extract treatment were considered as independent variables), followed by Šidák’s test in post hoc analysis. Results from further experiments with two-group comparisons were analysed by t-test with Welch’s correction for unequal variances. Data obtained from GFAP-positive staining experiments were not Gaussian distributed and were analysed by Mann–Whitney test. Data were expressed as mean + SEM (standard error of the mean) and statistical significance was set at *p* < 0.05, which was assessed in the GraphPad Prism 9.0.0 software. The total number of animals used in each experiment is described in figure legends.

## 5. Conclusions

Altogether, this study shows that CBD-enriched extracts administered from adolescence to early adulthood is not detrimental for the short-term memory of Wistar rats and offers no harm to locomotor activity in spite of the presence of THC in the *Cannabis* extract. Our data offers further results as, in addition to behavioural outputs, we evaluated neurochemical changes in the hippocampal circuits of adolescent rats treated with CBD. Indeed, changes in glutamatergic GluA1 AMPA receptor subunits and GFAP (reduction), or increased PSD95 levels might represent an important aspect to the dynamic allostasis of excitability of neuroglial circuits under the influence of CBD. Here, our study also showed a reduction in microglial complexity in the CA1 and CA3 regions, but no alterations in their phagocytic activity through Iba-1 and LAMP2 co-localization. It is imperative to gather data on the effects of phytocannabinoid exposure during the adolescent periods, in which the endocannabinoid system has a crucial role in neurodevelopmental processes [111] and thousands of children and adolescent have been recently treated. Collectively, our data reinforce the notion that CBD-enriched Cannabis treatment may be safe and well tolerated in healthy subjects, besides acting as a neuroprotective agent against hippocampal alterations related to the pathogenesis of excitatory and astrogliosis-mediated disorders in CNS.

## Figures and Tables

**Figure 1 ijms-24-11775-f001:**
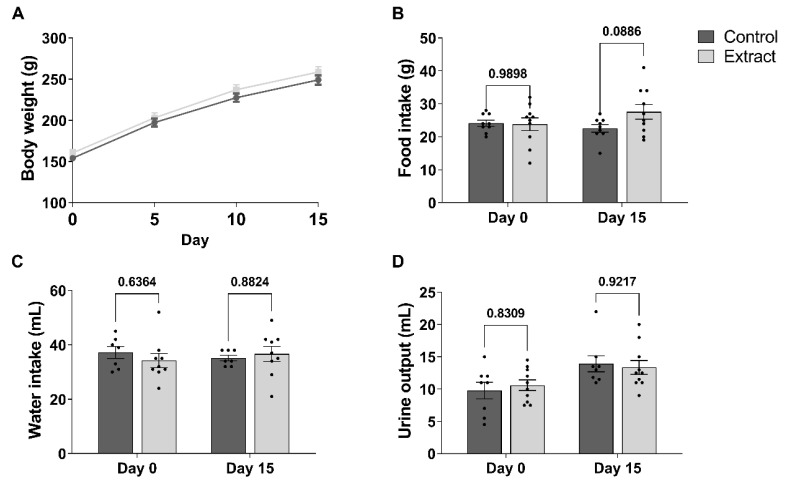
Administration of CBD-enriched extract does not affect weight gain, food and water intake, or urine flow in young animals. (**A**) Body weight gain measured before the start of treatment and every 5 days during its course. (**B**) Food intake, in grams, water intake in mL (**C**) and urine flow in mL/24 h (**D**) measured before (day 0) and after (day 15) treatment. Data were analysed by two-way ANOVA with Sidak’s correction for multiple comparisons and are presented as mean ± standard error of mean.

**Figure 2 ijms-24-11775-f002:**
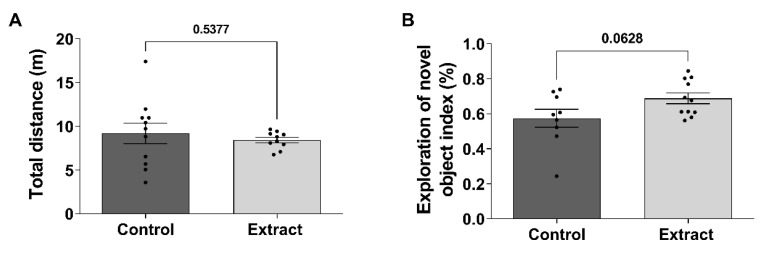
Treatment with CBD-enriched extract did not impair locomotor behaviour and cognitive performance. (**A**) Quantification of the total distance travelled during 5 min in the open field test. (**B**) Percentage of novel object exploration in short-term memory paradigm of NOR test. Data were analysed by the *t* test, with Welch’s correction for different standard deviations, and expressed as mean ± standard error of mean.

**Figure 3 ijms-24-11775-f003:**
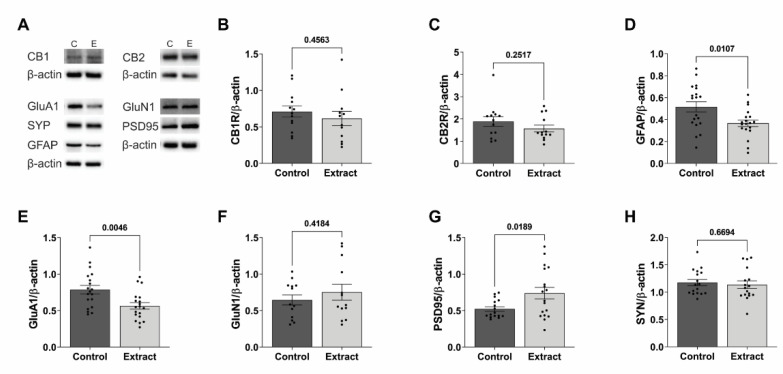
Treatment with CBD-enriched extract does not modulate the protein levels of cannabinoid receptors (CB1 or CB2), SYN or GluN1, but alters levels of GFAP, GluA1 and PSD95 in the hippocampus. (**A**) Representative images of Western blotting runs (β-actin was used as loading control and the respective band was depicted for different membranes). Relative protein levels of cannabinoid receptors, (**B**) CB1R and (**C**) CB2R, marker of astrogliosis GFAP (**D**), subunits of ionotropic glutamate receptors GluA1 (**E**) and GluN1 (**F**), and proteins associated to synaptic plasticity, notably PSD95 (**G**) and SYN (**H**). Data were analysed by the *t* test, with Welch’s correction for different standard deviations, and expressed as mean ± standard error of mean (*n* = 11–13).

**Figure 4 ijms-24-11775-f004:**
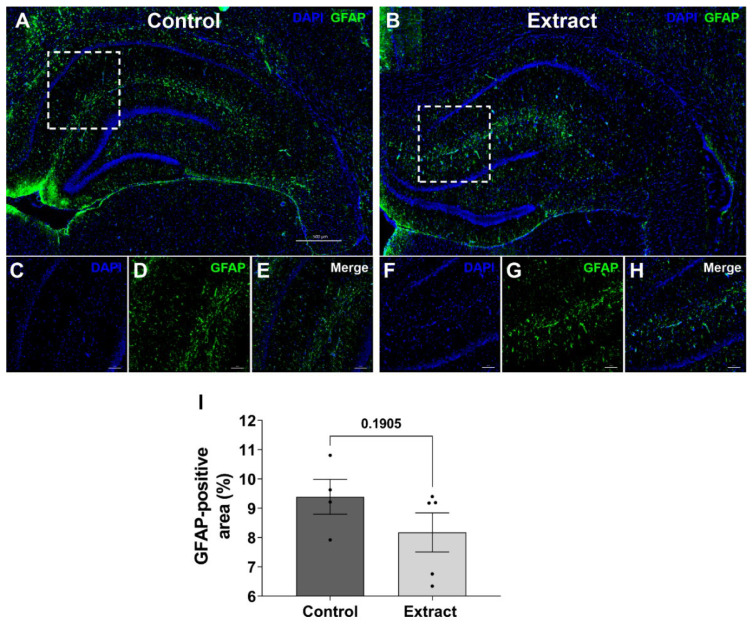
CBD-enriched extract did not modulate GFAP-positive staining in hippocampus. Representative images of immunohistochemistry for GFAP in (**A**) Control and (**B**) Extract group hippocampi at 20× magnification. CA1 regions (dashed boxes in **A** and **B**) were also highlighted 40× magnification (**C**–**H**). The total percent GFAP-positive area in hippocampus region has been quantified (**I**) and expressed in terms of mean ± standard error of mean, analysed by Mann–Whitney test (*n* = 4–5).

**Figure 5 ijms-24-11775-f005:**
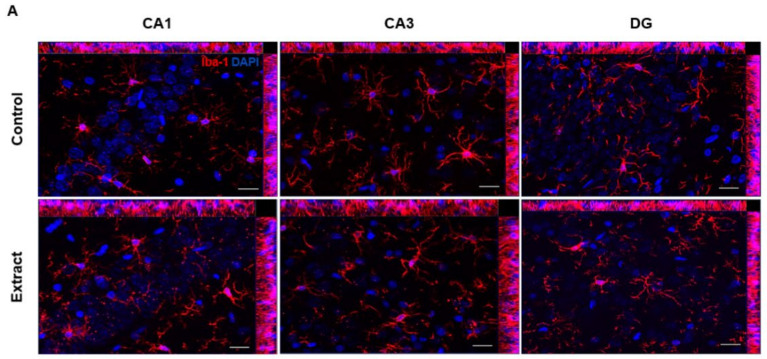
CBD-enriched extract reduces microglial complexity in CA1 and CA3 regions of hippocampus. Immunohistochemistry for identification of Iba-1 positive cells in CA1, CA3 and DG regions of hippocampus is representatively shown (**A**) at 40× magnification, and photomicrographs were taken in z-stack mode without predefined thickness, followed by Sholl analysis performed at ImageJ 1.53t software plugin (**B**). The total number of intersections in concentric circles, each 5 µm from soma, was quantified (**C**–**E**) and total area under curve was calculated for inference of microglia morphological complexity (**F**–**H**). Data from area under curve analysis were assessed by the *t* test, with Welch’s correction for different standard deviations, and expressed as mean ± standard error of mean (*n* = 3–4).

**Figure 6 ijms-24-11775-f006:**
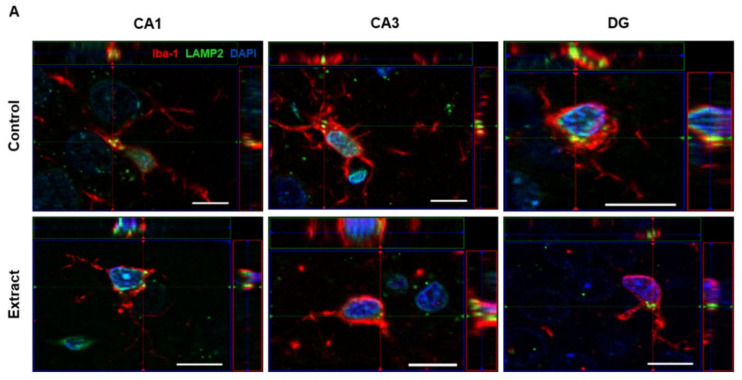
CBD-enriched Cannabis extract does not change the number of phagocytic events in microglia. (**A**) Photomicrographs of LAMP-2 and Iba-1 co-localization in CA1, CA3 and DG hippocampal regions in Control and Extract groups, with an increase of 63×. Total number of LAMP-2 positive puncta in microglial cells in CA1, CA3 and DG (**B**, **C** and **D**, respectively). Data were expressed as mean ± SEM and analysed by the *t* test, with Welch’s correction for different standard deviations (*n* = 5).

## Data Availability

The data presented in this study are openly available in FigShare at https://doi.org/10.6084/m9.figshare.23721159.v1 accessed on 17 July 2023.

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
