# Peer review of "Long-Term Treatment with Cannabidiol-Enriched Cannabis Extract Induces Synaptic Changes in the Adolescent Rat Hippocampus"

_ijms, 2023, doi:10.3390/ijms241411775_

Round 1

Reviewer 1 Report

In this work, Aguiar and colleagues evaluated the effect of cannabidiol on the hippocampus of adolescent rats.  The authors focused their analysis on physiological parameters and respective histological/protein analysis 15 days after the beginning f the treatment.  Overall, the authors showed no marked effects of cannabidiol on rat well-being and motor/memory performance.  The authors found significant changes on PSD95 levels (increase) and microglial levels in CA3 (decrease), whereas other parameters remained constant.  Overall the manuscript is well-written, the science behind it is solid and the conclusions are of interest for the overall scientific community.  I believe this manuscript can be accepted by IJMS, but some issues need to be solved by the authors.  Please return the manuscript to the authors with Major revisions.  Consider the following:

1. Please, check the phrase in lines 69-73 of the manuscript.  I believe there is information missing and/or a conclusion not completely taken.

2. In Figure 3, please change “β-actina” (Portuguese) to “β-actin” (English) in all graphics.

3. Please, consider the text in lines 213-216.  I believe there is missing information in this part of the manuscript.

4. Regarding Figure 6, what was the protocol used to quantify number of events? Were the cells counted manually or using a macro?  This information could be added to the materials and methods.

5. Considering the discussion section, lines 475-477.  Can the authors expand on cell phagocytic activity based on the immunohistochemistry results?

6.  Overall, the authors suggest that neural tissue plasticity may be improved with the administration of cannabidiol.  Can the authors expand whether they observed concomitant changes in oligodendrocyte maturations and/or neuron death/ramification.  I believe these are also some important changes that occur during brain development in adolescence, and as such would be of interest to complement the study.

7. Finally, can the authors comment on the possible long-term effects of cannabidiol administration in the adolescent rats, propose future research venues based on these results and translate these conclusions for human/therapeutic use of cannabidiol?

Reviewer 2 Report

(l. 20): Authors should report the term prior to the abbreviation "CBD".

(l. 21): Information about the age of animals when treatment was initiated should be included.

(l. 57, l. 78): :"express" (instead of "expresses") and "underlie"given that the subject is in plural.

(l. 92): "indicates"

(l. 506): Please report the dimensions of the cages.

(l. 507): The exact times (lights on/off) in the 12-hour light-dark cycle should be added.

l. 514): Please provide information regarding the oral administration (e.g., how was it conducted, what time of the day in terms of the circadian cycle?)

(l. 519-20): The exact group size (animals that were behaviorally tested and those housed in the metabolic cages) should be included.

(l. 522): Please report the time period of behavioral testing during the dark cycle as well as when it took place in terms of the drug administration.

(l. 525): Briefly mention the exact behavior(s) you wanted to study by administering both behavioral tests as well as the measures of each test that were analyzed for this purpose.

(l. 533; 536): Give some information about the type of objects and the criteria that had to be met in order a behavior to be defined as object exploration.

(l. 540): Were the animals euthanized the day after behavioral testing? Was anesthesia used?

(l. 563): It is implied that different animals were used for Western Blotting and immunohistochemistry. Please state this clearly and present the exact number of animals in each subgroup.

(l. 579): The coordinates of the hippocampal area from which sections were taken have to be reported.

(section 4.11): The dependent variables that were analyzed should be added.

(section 2.1): The time points of food intake, body weight measurements, water consumption and urine output have to be presented in the methods section.

(l. 141): The exact behaviors or tests should be reported instead of the expression "behavioural changes"

(l. 156): Object recognition is just a type of memory but it is well known that this type of non-spatial memory may be intact in contrast to other cognitive functions that may be more sensitive to specific experimental manipulations. Thus, the conclusion that based on the non-significant difference in the recognition index, there are no cognitive deficits, may be an oversimplification.

(sections 2.2, 2.3, 2.4, 2.5): statistical values are missing (e.g., degrees of freedom, t value, p). Presentation in most cases is limited to mean group values.

(l. 208): Authors state that "Considering the reduction observed in GFAP levels in total hippocampus homogenates evaluated by Western blotting, we further investigated the tissue distribution of GFAP-positive cells." However they analyzed only the CA1 subregion without exploring any other hippocampal subfields. Furthermore, authors state that the reason for conducting immunohistochemistry was the investigation of tissue-distribution of GFAP-positive cells given the reduced GFAP in total hippocampus homogenates. Based on this, it could be asked why authors did not do the same in the case of increased PSD-95.

Reviewer 3 Report

In this manuscript, Aguiar et al. evaluated the effect of CBD-enriched Cannabis extract on synaptic components in the hippocampus of rats from adolescence to early adulthood.

There are some concerns: 

1) Sample size and statistical power: The study does not specify the exact number of rats used in each experimental group. Without this information, it is difficult to assess the statistical power and reliability of the results. A larger sample size would increase confidence in the findings and improve the ability to detect meaningful effects.

2) Lack of behavioral tests for animals used in metabolic cage assessment: The animals used for measuring water and food intake, as well as urine flow, in the metabolic cages were not subjected to behavioral tests. This means that the effects of the extract on behavior, which could potentially impact the animals' metabolic parameters, were not assessed in this specific group.

3) Limited scope of behavioral tests: The study only included two behavioral tests, namely the open field test and the novel object recognition (NOR) test. While these tests provide valuable insights into general locomotor activity and recognition memory, they do not comprehensively assess the animals' cognitive abilities or emotional behavior. Including a broader range of behavioral tests would provide a more comprehensive evaluation of the extract's effects. Other behavioral tests like Morris Water Maze, Elevated Plus Maze, Fear Conditioning can be considered.

4) Potential confounding factors: The study does not provide information about potential confounding factors that might influence the results. Factors such as circadian rhythm, stress, and habituation to testing procedures could impact the animals' behavior and potentially confound the interpretation of the results. Controlling for and reporting on these factors would enhance the validity of the findings.

Minor English editing is required. 

Round 2

Reviewer 1 Report

In this work, Aguiar and colleagues evaluated the effect of cannabidiol on the hippocampus of adolescent rats.  The authors focused their analysis on physiological parameters and respective histological/protein analysis 15 days after the beginning f the treatment.  Overall, the authors showed no marked effects of cannabidiol on rat well-being and motor/memory performance.  The authors found significant changes on PSD95 levels (increase) and microglial levels in CA3 (decrease), whereas other parameters remained constant.  Overall the manuscript is well-written, the science behind it is solid and the conclusions are of interest for the overall scientific community.

Most of my questions have been answered satisfactorily, and the manuscript has been improved in the process.  I believe the manuscript can be accepted for publication, but only after the authors address a final minor comment.

1. The information provided in your answers to my comments 6 and 7 are valuable and should be added to the manuscript.  In particular, the text used to answer to comment 7 appears to be the perfect conclusion for this manuscript: it combines perfectly the conclusions of the study with available literature and potential translational applications.  Consider adding both texts to the manuscript, I believe it will value it even more.

Reviewer 2 Report

l. 546: "there was no administration of treatment": The expression is not clear. I assume you mean that no drug administration was performed. Please rephrase.

Sections 4.5 & 4.6: Authors should add references regarding the behavioral tests and the evaluation process (e.g., measurements to estimate specific behaviors).

l. 604: authors should add the rat atlas they used.

Regarding the presentation of the statistics, authors have stated that they added degrees of freedom, as well as t and p values. However, the presentation is not appropriate. For example in l. 156, statistics should appear as follows t (18) =1,983,  p= 0.062. Please make appropriate corrections  in all data.

Regarding last comment (referring to l. 208 of the initial submission), authors should incorporate their answer into the revised manuscript.

Reviewer 3 Report

The authors justified their concerns appropriately, and I feel the paper is good for publication.
